# Estimated 24 h Urinary Sodium-to-Potassium Ratio Is Related to Renal Function Decline: A 6-Year Cohort Study of Japanese Urban Residents

**DOI:** 10.3390/ijerph17165811

**Published:** 2020-08-11

**Authors:** Hiroko Hattori, Aya Hirata, Sachimi Kubo, Yoko Nishida, Miki Nozawa, Kuniko Kawamura, Takumi Hirata, Yoshimi Kubota, Mizuki Sata, Kazuyo Kuwabara, Aya Higashiyama, Aya Kadota, Daisuke Sugiyama, Naomi Miyamatsu, Yoshihiro Miyamoto, Tomonori Okamura

**Affiliations:** 1Graduate School of Health management, Keio University, Kanagawa 252-0883, Japan; ymsaht@gmail.com; 2Department of Health and Nutrition, Tokiwa University of Human Science, Ibaraki 310-8585, Japan; 3Department of Preventive Medicine and Public Health, Keio University School of Medicine, Tokyo 160-8582, Japan; aya.hirata@keio.jp (A.H.); msata@keio.jp (M.S.); kuwabara@a6.keio.jp (K.K.); dsugiyama@z8.keio.jp (D.S.); 4Foundation for Biomedical Research and Innovation at Kobe, Kobe, Hyogo 650-0047, Japan; sachimik@gmail.com (S.K.); nishida-kndn@umin.org (Y.N.); kawamura@fbri.org (K.K.); t-hirata@med.hokudai.ac.jp (T.H.); yo-kubota@hyo-med.ac.jp (Y.K.); ayahiga@wakayama-med.ac.jp (A.H.); ayakd@belle.shiga-med.ac.jp (A.K.); miyan@belle.shiga-med.ac.jp (N.M.); miyamoty@ncvc.go.jp (Y.M.); 5Faculty of Human Sciences, Tezukayama Gakuin University, Osaka 590-0113, Japan; 6Japan Health Insurance Association, Saitama 330-8686, Japan; miki-n@keio.jp; 7Department of Public Health, Hokkaido University Faculty of Medicine, Hokkaido 060-8638, Japan; 8Department of Environmental and Preventive Medicine, Hyogo College of Medicine, Hyogo 663-8501, Japan; 9Department of Hygiene, Wakayama Medical University, Wakayama 641-8509, Japan; 10Center for Epidemiologic Research in Asia Shiga University of Medical Science, Shiga 520-2192, Japan; 11Faculty of Nursing and Medical Care, Keio University, Kanagawa 252-0883, Japan; 12Department of Clinical Nursing, Shiga University of Medical Science, Shiga 520-2192, Japan; 13Open Innovation Center, National Cerebral and Cardiovascular Center, Osaka 564-8565, Japan

**Keywords:** urinary sodium-potassium ratio, urinary sodium, estimated GFR, renal function

## Abstract

The effect of the sodium-to-potassium ratio (Na/K) on renal function within the clinically normal range of renal function are limited. We investigated the effects of an estimated 24 h urinary Na/K (e24hUNa/K) on a 6-year renal function decline among 927 urban Japanese community dwellers with no history of cardiovascular diseases and medication for hypertension, diabetes, or dyslipidemia. We partitioned the subjects into quartiles according to the e24hUNa/K. The estimated glomerular filtration rate (eGFR) was calculated using the chronic kidney disease epidemiology collaboration (CKD/EPI) formula and renal function decline was defined as an absolute value at or above the third quartile of the eGFR decline rate. A multivariable logistic regression model was used for estimation. Compared with the first quartile of the e24hUNa/K, multivariable-adjusted odds ratios (ORs) for eGFR decline in the second, third, and fourth quartiles were 0.96 (95% confidence interval: 0.61–1.51), 1.06 (0.67–1.66), and 1.65 (1.06–2.57), respectively. These results were similar when the simple spot urine Na/K ratio was used in place of the e24hUNa/K. Apparently healthy urban residents with an almost within normal range mean baseline eGFR and high e24hUNa/K ratios had an increased risk for a future decline in renal function. Reducing the Na/K ratio may be important in the prevention of chronic kidney disease in its early stage.

## 1. Introduction

The number of patients with chronic kidney disease (CKD) continues to rise due to population ageing. CKD is a critical global health concern because it is a major risk factor for end-stage renal disease and cardiovascular diseases [1,2,3]. The examination of the risk factors for age-related kidney function decline could help in understanding the mechanisms of kidney ageing. CKD progression can be reduced by improving dietary habits and receiving appropriate treatment from an early stage [4,5]. In a large group of normotensive and never-treated patients with essential hypertension an important role for sodium intake on the interaction between systolic arterial pressure and albuminuria was demonstrated. High salt intake increased the excretion of protein in the urine, resulting in a decrease in renal function [6]. Increased intake of potassium increases the excretion of sodium in the urine and decreases blood pressure and reducing kidney damage [7]. Recently, it was reported that a high dietary sodium-to-potassium ratio (Na/K) was associated with increased risk of CKD incidence [8] and a high 24 h urinary Na/K ratio was also associated with CKD progression [9]. Twenty-four hour urinary sodium excretion [10], salt intake [11], and low urinary potassium excretion [12] were associated with CKD progression. However, relevant findings on the effect of the Na/K ratio on the development of CKD are still limited [8,9,13,14]; furthermore, although estimated glomerular filtration rate (eGFR) and proteinuria are insensitive indicators for predicting renal function decline, and future development of CKD, little is known about the association between Na/K ratio and renal function in a population with a clinically normal range of renal function.

Therefore, this study aimed to investigate the effects of an estimated 24 h urinary sodium-to-potassium ratio (e24hUNa/K) on renal function decline among apparently healthy urban residents in Japan.

## 2. Materials and Method

### 2.1. Study Participants

This was a 6-year follow-up study of the Kobe Orthopedic and Biomedical Epidemiological (KOBE) study, a population-based cohort study of citizens living in Kobe City. Detailed information about the KOBE study has been previously published [15,16,17,18,19]. In a nutshell, 1117 residents of Kobe City were recruited between July 2010 and December 2011 through Kobe City’s website and public relations. The inclusion criteria were as follows: (1) Age 40 to 74 years, (2) no history of malignant neoplasm or cerebral or cardiovascular disease, (3) not on medication for hypertension, diabetes, or dyslipidemia, (4) subjectively healthy, (5) having the ability to arrive at the study site for requisite investigations, and (6) consent to participation in follow-up studies. Research participants were invited for an onsite survey every two years. The survey was done with a 2-year interval for each participants because we recruited apparently healthy community residents, of which examination cycle is as follows: Baseline survey (2010–2011), Follow-up 1 (2012–2013), Follow-up 2 (2014–2015), Follow-up 3 (2016–2017), Follow-up 4 (2018–2019).

This study is a follow-up study from the baseline survey to the 6-year follow-up survey. A number of participants were excluded for the following reasons: Missing data in the baseline survey (*n* = 2), incomplete follow-up (*n* = 186), and missing data in the 6-year survey (*n* = 2). Finally, 927 participants (282 men and 645 women) were included in this study. The participant flow diagram is shown in Figure 1.

### 2.2. Measurements

Participants completed a standardized questionnaire of their medical history and lifestyle habits (e.g., smoking, alcohol consumption, and physical activity), and trained researchers confirmed the response contents through face-to-face communications. The height and weight of the participants in socks and light clothing were measured using a combined meter (U-WELL2; Elk Corp, Oosaka, Japan). Body mass index (BMI) was calculated as weight (kg) divided by squared height (m^2^). Blood pressure measurements were taken twice with an automatic sphygmomanometer (BP-103i II; Nihon Colin, Tokyo, Japan) after participants were allowed to rest in a seated position for at least 5 min (measured by hourglass), and the mean value was recorded.

Blood samples were collected after ≥10 h of fasting, and all blood samples from the participants were assessed by the largest designated laboratory (SRL Inc., the largest clinical laboratory in Japan; https://www.srl-group.co.jp/english/). Blood glucose levels (mg/dL) were measured using the glucose oxidase method and hemoglobin A1c was assessed using the latex agglutination test. Total cholesterol, high-density lipoprotein cholesterol (HDL-C), and triglycerides (TG) levels were measured using enzymatic methods, and low-density lipoprotein cholesterol (LDL-C) levels were calculated using the Friedewald equation [20]. Serum high-sensitivity C-reactive protein (hs-CRP) was measured using a BN II nephelometer (Dade Behring, Deerfield, IL, USA) as previously reported [18].

Urine samples were collected at the research site in the morning without breakfast. The collected fresh urine was taken in a clean urine test cup and immediately dispensed into a sterile tube and was under refrigeration before measurement. The urine albumin level was measured by turbidimetric immunoassay. The urine creatinine level was measured using the enzymatic method. Urine sodium and potassium levels were measured using the electrode method.

The e24hUNa/K was defined as the ratio of estimated 24 h urinary sodium to estimated 24 h urinary potassium as calculated using a spot urine by Tanaka et al.’s equations [21]. The equations were as follows:PRCr (mg/day) = −2.04 × age + 14.89 × weight (kg) + 16.14 × height (cm) − 2244.45;(1)
Estimated 24 h urinary sodium (e24hUNa) (mEq/day) = 21.98 × XNa^0.392^;(2)
Estimated 24 h urinary potassium (e24hUK) (mEq/day) = 7.59 × XK^0.431^;(3)
Estimated 24 h Na and K excretion (e24hUNa/K) = e24hUNaV/e24hUKV.(4)
where PRCr = predicted value of 24hUCr, SUNa = Na, SUK = K concentration in the spot voiding urine, SUCr = creatinine concentration in the spot voiding urine, XNa (or XK) = SUNa (or SUK)/(SUCr × 10) × PRCr.

We partitioned the subjects into quartiles according to the 24 h urinary Na/K ratio. (Q1: e24hUNa/K < 2.8, Q2: 2.8 ≤ e24hUNa/K < 3.2, Q3: 3.2 ≤ e24hUNa/K < 3.6, Q4: e24hUNa/K ≥ 3.6.)

Additional analysis using simple urinary Na/K ratio (UNa/K) was also performed.

### 2.3. Renal Function Decline

Serum creatinine was also measured in SRL Inc. using the enzymatic method. Our serum creatine values were not IDMS-traceable, but based upon acid metric titration method primarily calibrated by reference standard NIST SRM914. This reference standard is also used for primary calibration for the reference material JCCRM 521 which is determined by IDMS. Moreover, serum creatine is evaluated by external accuracy audit in Japan; the intra and inter method coefficient of variation for serum were 3.45% and 5.60%, respectively.

The estimated glomerular filtration rate (eGFR) was calculated using the serum creatinine from serum samples, collected at the time of the survey, by the following CKD/EPI formula [22]:eGFR = 141 × min (Scr/κ,1) ^α^ × max (Scr/κ,1)^−1.209^ × 0.993^age^ × 1.018 [if female] × 0.813 [if Japanese](5)

In this equation, Scr is serum creatine in mg/dL; κ is 0.7 and 0.9 for men and women, respectively; α is −0.329 and −0.411 for men and women, respectively; min indicates the minimum of Scr/κ or 1, and max indicates maximum of Scr/κ or 1.

The rate of eGFR decline from baseline to 6 years was divided into quartiles for all participants. We defined eGFR decline as the decline value at or above the third quartile of the decline rate (−8.02% during 6 years = −1.34%/year).

### 2.4. Statistical Analysis

Analysis of variance and the chi-squared test were used to estimate the means and the prevalence of baseline characteristics. The multivariable-adjusted odds ratios (ORs) and 95% confidence intervals (CI) of each quartile group of e24hUNa/K for eGFR decline compared with the first quartile group were estimated using a multivariable logistic regression model. Adjusted variables were age, BMI, alcohol consumption (current drinking, past drinking, and no drinking), smoking (current smoking, past smoking, and no smoking), HbA1c, HDL-C, LDL-C, eGFR (baseline), and hypertension (baseline: SBP ≥ 130 or/and DBP ≥ 80) [23]. There was no interaction between men and women regarding the relationship between e24hUNa/K and renal function decline; thus, sex-combined analysis with adjustment for sex was performed in the multivariable analysis. All the analyses were repeated using UNa/K in place of e24hUNa/K.

Statistical analysis was performed using IBM SPSS Statistics version 25 (IBM Corp., Armonk, NY, USA), with a two-tailed 5% level of significance.

### 2.5. Statement of Ethics

This research has been approved by the Pharmaceutical Clinical Research Review Committee (Ethics Committee) of the Institute of Biomedical Research and Innovation at the Kobe Biomedical Innovation Cluster (approval no. 10–20) and the Ethics Committee of Keio University School of Medicine (approval no. 20170142). Participants were given written and oral explanations, and written informed consent was obtained.

## 3. Results

In the baseline survey, means (standard deviations (SDs)) for e24hUNa/K, e24hUNa, and e24hUK were 3.2(0.7), 145(32) mEq/day, and 46(8) mEq/day, respectively. The mean (SD, min–max) eGFR was 79.2 (8.0, 48.8–104.8) mL/min/1.73 m^2^.

### 3.1. Baseline Characteristics of Study Participants According to the Quartile Groups of e24hUNa/K Levels

Table 1 shows participant characteristics according to the e24hUNa/K quartile group. Waist circumference, BMI, systolic blood pressure (SBP), diastolic blood pressure (DBP), and prevalence of hypertension were lower in the lowest Na/K group (Q1) than in the other groups (Q2–4). HDL-C was higher in the lowest group (Q1) than the other groups (Q2–4).

### 3.2. Multivariable Adjusted Means of eGFR and Its 6-Year Change According to the Quartile Groups of e24hUNa/K Levels

Table 2 shows the multivariable-adjusted means of the e24hUNa/K quartile group for the eGFR-decrease during the 6-year period. Means of absolute amount of change in eGFR were higher in higher e24hUNa/K quartiles (Q1: −0.72 (95% CI: −0.81–−0.64), Q2: −0.75 (−0.84–−0.67), Q3: −0.81 (−0.90–−0.73), Q4: −0.96 (−1.05–−0.87) mL/min/1.73 m^2^/year) (*p* = 0.001). The same was in the means of absolute change rate in eGFR (Q1: −0.91 (95% CI: −1.03–−0.78), Q2: −0.95 (−1.07–−0.83), Q3: −1.04 (−1.16–−0.92), Q4: −1.22 (−1.34–−1.10) %/year) (*p* = 0.002). These results were similar according to the UNa/K quartile group (Appendix A).

### 3.3. Multivariable-Adjusted Odds Ratio for eGFR Decrease According to the Quartile Groups of e24hUNa/K Levels

Figure 2 shows the multivariable-adjusted OR for the eGFR decrease according to the e24hUNa/K quartile group. Compared with Q1, ORs (95% CI) for eGFR decline were as follows: Q2: 0.96 (0.61–1.51); Q3: 1.06 (0.67–1.66); and Q4: 1.65 (1.06–2.57), respectively. These results were similar according to the UNa/K quartile group (Appendix A). Furthermore, the above-mentioned findings were not substantially affected when a 10% decrease of eGFR in each participant (mean absolute decline −15.1%) was set as an outcome (Appendix A).

Urinary albumin was measured at baseline, and we defined being 300 mg/g·Cre or more as overt proteinuria (*n* = 2), both of which were in menstruating women. In the follow-up survey 6 years later, a urine qualitative test was conducted by a dipstick test not by urinary albumin; and ≥1+ on dipstick was defined as proteinuria (*n* = 16). However, the results of Figure 2 (Appendix A) did not change even if these subjects were excluded (*n* = 18). Further adjustment for baseline hs-CRP (log-transformed) did not also alter the results (date not shown).

## 4. Discussion

In the present study, the mean eGFR decrease was 0.8 mL/min/1.73 m^2^/year, which was similar to those of previous studies [24,25,26]; and during the 6-year follow-up period, increased ORs for eGFR decline were observed with higher 24hUNa/K ratios in the apparently healthy urban residents with mean baseline eGFR of about 80 mL/min/1.73 m^2^, who had no past history of neoplasms, cardiovascular diseases, or medication for hypertension, diabetes, or dyslipidemia. There were no participants with stage 3b or more of CKD (eGFR < 45 mL/min/1.73 m^2^). Similar results were observed when UNa/K was used instead of e24hUNa/K in our statistical models. The present study demonstrated that higher e24hUNa/K levels were associated with a higher risk of eGFR decline within an almost normal range of eGFR. This finding suggests that increased intake of potassium in addition to reduced intake of sodium might prevent renal function decline in the early stage of CKD.

A prospective cohort study of Iranian adults that estimated sodium and potassium levels using a food frequency questionnaire examined the risk of CKD within a 6.3-year period [8]; participants in the highest tertile of the Na/K ratio had a higher OR of CKD (OR = 1.52) than those in the lowest tertile. Furthermore, the researchers found out that sodium and potassium alone were not associated with a risk of CKD. The Korean N Cohort Study of Outcomes in Patients with CKD (KNOW-CKD) was analyzed in a 5-year prospective study of 1001 patients with non-dialysis-dependent CKD with a mean baseline eGFR of about 50 mL/min/1.73 m^2^ [9]. Subjects were divided into quartiles according to 24 h urinary Na/K ratio. Compared to the lowest quartile of 24 h urinary Na/K ratio, the odds ratio for renal outcomes were 2.48 (95% CI 1.30–4.90) in the third quartile and 2.95 (95% CI 1.56–5.81) in the fourth quartile, respectively.

In the previous studies, 24 h urinary sodium excretion [10] or salt intake and eGFR based on a food frequency questionnaire [11] were associated with CKD progression. Regarding the association between dietary potassium intake and CKD, a cohort study involving 5315 Dutch men and women reported that low urinary potassium excretion increased the risk of CKD onset 5 years later [12]. Consequently, the combination of sodium and potassium such as e24hUNa/K or 24 h urinary Na/K ratio is supposed to be a useful marker for future decline in renal function. Our study also suggested that urinary Na/K ratio may be a simple marker for this purpose.

The following mechanism is considered to be a factor to explain the relationship between the sodium–potassium ratio and blood pressure. A large amount of sodium intake causes cardiac dysfunction and renal dysfunction, and it is known that the main mechanism is the enhancement of oxidative stress, which enhances insulin resistance [27]. On the contrary, potassium has an antioxidant effect at the same time as a natriuretic effect, and it is known in animal experiments that potassium administration improves insulin resistance due to sodium [28]. It has also been suggested that there is a common mechanism in the process of formation of salt sensitivity and insulin sensitivity of blood pressure [29].

High sodium intake and low potassium intake are common dietary problems in Japan. In the 2018 National Health and Nutrition Survey, the average daily sodium intake (salt equivalent) measured by the one-day weighing method in Japanese subjects aged 20 years or older was 11.0 g per day for men and 9.3 g per day for women. In 2012, the WHO proposed a recommended potassium intake of 3510 mg per day for hypertension prevention in adults [30]; yet the average potassium intake was only 2386 mg per day for men and 2205 mg per day for women [31]. The results of the International Study of Macro-/Micro-nutrients and Blood Pressure (INTERMAP Study) [32] from a four-country collaborative study including Japan, China, the United Kingdom, and the United States showed that sodium excretions measured by 24 h urine collections were low at 2929–4202 mg/day (salt equivalent 7.4–10.7 g/day) in Westerners and high at 4843 mg/day (salt equivalent 12.3 g/day) in men and at 4278 mg/day (salt equivalent 10.9 g/day) in Japanese women. Conversely, potassium was high in Western populations, ranging from 1982–2912 mg/day, and low in Japanese populations, at 1920 mg/day in men and 1891 mg/day in women. The Na/K ratio calculated by 24 h urine collections also tended to be low at 2.2–3.1 in Western populations, and high for Japanese subjects at 4.5 in men and 4.1 in women. Therefore, lifestyle modification guidance for the prevention of renal function decline and focusing on the Na/K ratio may be particularly helpful in the Japanese population.

This study had several limitations. First, the KOBE study consisted of subjectively healthy participants from an urban area who voluntarily participated in the survey. Therefore, caution is required when applying the results of this study to other populations. Indeed, even compared to Japanese workers in a study that used the same estimation formula to measure e24hUNa and e24hUK (HIPOP-OHP study) [33,34,35], participants in the present study had higher e24hUK and lower e24hUNa and BMI. Second, a 24 h urine collection is recommended for accurate measurement of salt intake, but estimations were calculated from spot urine for participants in the present study. However, this method is considered to be capable of estimating salt excretion to a certain extent at least at the general population level [36]. Third, in this study, we were not able to examine whether nutrients other than sodium and potassium influenced the results because a detailed nutrition survey was not conducted at baseline. Fourth, the participants of this survey are completely healthy participants, who cannot follow the course of eGFR on a monthly basis like hospital patients, and it is predicted that few patients will have renal impairment in the first few years. Therefore, it was designed like this research. A study design that can more closely track the eGFR of healthy participants is needed, and this is a future subject.

## 5. Conclusions

In conclusion, individuals with high e24hUNa/K ratios had an increased risk for a future decline in renal function in the apparently healthy urban Japanese residents, who had a mean baseline eGFR almost within the normal range and had no past history of cardiovascular diseases, neoplasms, or medication for hypertension, diabetes, or dyslipidemia. These findings suggest that reducing the Na/K ratio is important in the prevention of the progression of renal function decline in its early stage. A dietary guidance focusing on the Na/K ratio may be a useful first step in the prevention of CKD.

## Figures and Tables

**Figure 1 ijerph-17-05811-f001:**
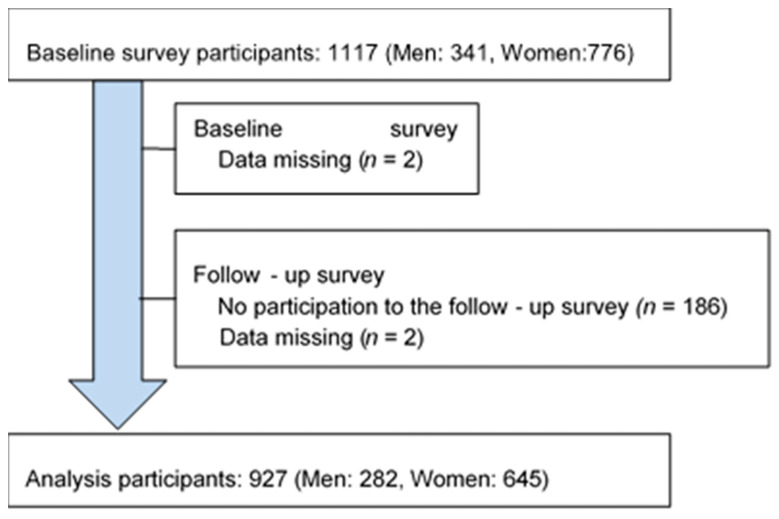
Study participants.

**Figure 2 ijerph-17-05811-f002:**
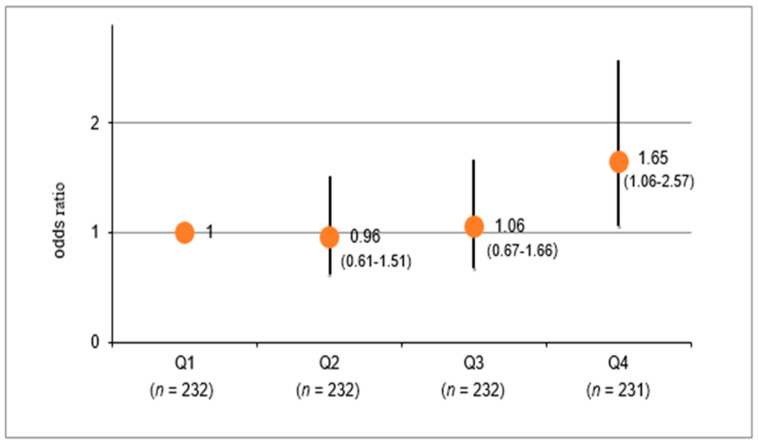
Multivariable-adjusted odds ratio for eGFR decrease according to the quartile groups of e24hUNa/K levels. Q1: e24hUNa/K < 2.8, Q2: 2.8 ≤ e24hUNa/K < 3.2, Q3: 3.2 ≤ e24hUNa/K < 3.6, Q4: e24hUNa/K ≥ 3.6; Data are odds ratio and 95% confidence interval. Logistic regression models were used; Adjusted for sex, age, BMI, cigarette smoking (current/past/none), alcohol drinking (current/past/none), HDL-C, LDL-C, HbA1c, eGFR (baseline), hypertension; eGFR: Estimated glomerular filtration rate; BMI: Body mass index, HDL-C: High-density lipoprotein cholesterol; LDL-C: Low-density lipoprotein cholesterol; Hypertension: SBP ≥ 130 or/and DBP ≥ 80.

**Table 1 ijerph-17-05811-t001:** Baseline characteristics of study participants according to the quartile groups of e24hUNa/K levels.

Characteristics	Total		Q1		Q2		Q3		Q4		*p* for Trend
Number	927		232		232		232		231		
Age, years	58.9	(8.6)	58.2	(9.1)	59.3	(8.6)	58.8	(8.5)	59.3	(8.0)	0.43
Women	645	(69.6)	174	(75.0)	161	(69.4)	163	(70.3)	147	(63.6)	0.07
Waist circumference, cm	79.7	(8.4)	78.3	(8.3)	79.9	(8.4)	80.3	(8.4)	80.4	(8.4)	0.03
BMI, kg/m^2^	21.5	(2.8)	21.1	(2.7)	21.5	(2.8)	21.8	(2.7)	21.8	(3.0)	0.01
Current smoking	40	(4.3)	12	(5.2)	6	(2.6)	8	(3.4)	14	(6.1)	0.24
Current drinking	473	(51.0)	111	(47.8)	115	(49.6)	119	(51.3)	128	(55.4)	0.40
SBP, mmHg	116.3	(17.4)	111.6	(15.2)	117.7	(17.3)	116.3	(17.3)	119.7	(18.9)	<0.001
DBP, mmHg	72.0	(11.1)	69.2	(9.9)	73.5	(10.9)	71.5	(11.5)	73.8	(11.3)	<0.001
Hypertension	274	(29.6)	42	(18.1)	79	(34.1)	69	(29.7)	84	(36.4)	<0.001
HbA1c, %	5.2	(0.4)	5.2	(0.4)	5.2	(0.6)	5.2	(0.4)	5.2	(0.4)	0.98
Glucose, mg/dL	89.8	(11.7)	89.3	(8.6)	90.2	(17.6)	89.7	(8.7)	90.0	(9.2)	0.85
HDL-C, mg/dL	68.6	(16.3)	70.8	(17.1)	68.8	(16.3)	69.0	(15.1)	65.8	(16.3)	0.01
LDL-C, mg/dL	131.0	(28.2)	129.6	(28.6)	132.3	(28.3)	131.5	(28.0)	130.5	(27.9)	0.76
TG, mg/dL	74.0 (55.0, 104.0)		69.0 (52.0, 96.8)		76.0 (56.0, 98.0)		72.0 (55.3, 105.5)		80.0 (57.0, 116.0)		0.02
hs-CRP, mL/L	225.0 (184.0, 480.0)		179.5 (93.9, 475.8)		248.5 (104.3, 509.0)		226.5 (120.0, 421.0)		232.0 (101.0, 469.0)		0.27
e24hUNa, mEq/day	144.6	(32.2)	117.4	(25.4)	138.7	(23.4)	153.9	(26.2)	168.7	(28.9)	<0.001
e24hUK, mEq/day	45.9	(8.1)	48.9	(8.8)	46.8	(7.7)	45.9	(7.6)	41.9	(6.8)	<0.001
e24hUNa/K	3.2	(0.7)	2.4	(0.3)	3.0	(0.1)	3.4	(0.1)	4.0	(0.4)	<0.001
e24hUsalt, g/day	8.5	(1.9)	6.9	(1.5)	8.2	(1.4)	9.1	(1.5)	9.9	(1.7)	<0.001
e24hUK, mg/day	1794	(318)	1910	(343)	1830	(300)	1794	(298)	1640	(268)	<0.001
ACR, mg/g·Cre	8.6 (5.7, 13.8)		8.4 (5.7, 12.4)		7.9 (5.8, 13.8)		8.7 (5.5, 13.4)		9.5 (5.8, 17.1)		0.32
eGFR, mL/min/1.73 m^2^	79.2	(8.0)	78.6	(8.1)	78.5	(8.8)	79.7	(7.6)	80.2	(7.1)	0.07

Q1: e24hUNa/K < 2.8, Q2: 2.8 ≤ e24hUNa/K < 3.2 Q3: 3.2 ≤ e24hUNa/K < 3.6 Q4: e24hUNa/K ≥ 3.6; Continuous data were analyzed using Student’s *t*-test, and were shown in the mean (standard deviation). Values of TG, hs-CRP, and ACR are the median (inter-quartile range); Categorical data were analyzed using the χ^2^ test, and were shown as number (%); BMI: Body mass index, SBP: Systolic blood pressure, DBP: Diastolic blood pressure, hypertension: SBP ≥ 130 or/and DBP ≥ 80; HDL-C: High-density lipoprotein cholesterol, LDL-C: Low-density lipoprotein cholesterol, TG: Triglyceride, hs-CPR: High-sensitivity C-reactive protein; e24hUNa: Estimated 24 h urine sodium excretion, e24hUK: Estimated 24 h urinary potassium excretion; e24hUNa/K: Estimated 24 h urine sodium-potassium ratio, e24hUsalt: Estimated 24 h urinary salt excretion. Salt equivalent (g) = Na (mEq) × molecular weight 23.0 × 2.54 (Na 23.0/NaCl 58.4) ÷ 1000, ACR: Albumin creatinine ratio; eGFR: Estimated glomerular filtration rate.

**Table 2 ijerph-17-05811-t002:** Multivariable-adjusted means of estimated glomerular filtration rate (eGFR) and its 6-year change according to the quartile groups of e24hUNa/K levels.

	Q1 (*n* = 232)		Q2 (*n* = 232)		Q3 (*n* = 232)		Q4 (*n* = 231)		*p*-Value
	Mean	(95% CI)	Mean	(95% CI)	Mean	(95% CI)	Mean	(95% CI)	
eGFR (6 years later), mL/min/1.73 m^2^	74.9	(74.4–75.4)	74.7	(74.2–75.2)	74.4	(73.8–74.9)	73.5	(73.0–74.0)	0.001
eGFR amount of change, mL/min/1.73 m^2^/year	−0.72	(−0.81–−0.64)	−0.75	(−0.84–−0.67)	−0.81	(−0.90–−0.73)	−0.96	(−1.05–−0.87)	0.001
change rate in eGFR%/year	−0.91	(−1.03–−0.78)	−0.95	(−1.07–−0.83)	−1.04	(−1.16–−0.92)	−1.22	(−1.34–−1.10)	0.002

Q1: e24hUNa/K < 2.8, Q2: 2.8 ≤ e24hUNa/K < 3.2, Q3: 3.2 ≤ e24hUNa/K < 3.6, Q4: e24hUNa/K ≥ 3.6; Data are presented as mean and 95% confidence interval. Analysis of variance were used to adjust for sex, age, BMI, cigarettes smoking (current/past/none), alcohol drinking; (current/past/none), HDL-C, LDL-C, HbA1c, eGFR (baseline), hypertension; *p*-Values for difference between groups; eGFR: estimated glomerular filtration rate, BMI: Body mass index, HDL-C: High-density lipoprotein cholesterol, LDL-C: Low-density lipoprotein cholesterol; hypertension: SBP ≥ 130 or/and DBP ≥ 80.

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
