# Peer review of "Estimated 24 h Urinary Sodium-to-Potassium Ratio Is Related to Renal Function Decline: A 6-Year Cohort Study of Japanese Urban Residents"

_ijerph, 2020, doi:10.3390/ijerph17165811_

Round 1
Reviewer 1 Report
This is an interesting study from Japan. Thanks for the invitation as a reviewer.These studies aimed to investigate the effects of an estimated 24-h urinary sodium-to-potassium ratio (e24hUNa/K) on renal function decline among apparently healthy urban residents in Japan. The studio seemed well-designed and structured. I have a number of concerns that should be addressed before publishing:
Introduction:
1:The authors must address that estimated glomerular filtration rate and proteinuria both are insensitive predictors of future decline in kidney function. Therefore, simple and alternative biomarkers are extremely interesting in these field.
2: Are there other references that can back up reference number 7? Add if possible
Material and Method, Statistical Analysis & Results
3: Missing: Detailed description of how follow-up has been carried out. Is it annual?
4: Missing: How and when was the urine sample collected? Please describe in detail
5: Missing: How are Na + and K + in the urine analyzed? Describe the method.
6: Line 112: Is an IDMS-traceable creatinine enzymatic assay used.
7: Line 112: What is the coefficient of variation on the method of analysis for creatinine?
8: Line 120: Will it be possible to add an analysis where you calculate the relative risk of a 10% decrease in eGFR within eg 1, 3 and 6 years between the quartiles. It will increase the clinical relevance and understanding of the study (major).
9: Do you have the possibility to adjust for proteinuria and CRP? In that case, it must be done and will increase the quality of the study. If not, it should be mentioned in the discussion under limitations of the study (major).
10: Please also present the unadjusted results. This will again increase the understanding and intelligibility (major).
Discussion:
12: Line 177: could be 0.8 mL/min/1.73m2/year.
13: Is it possible to briefly describe the mechanisms for why high Na + and low K + have a negative impact on kidney function over time.
Conclusion:
Nihil.
Author Response
Thank you very much for your valuable comments.We have revise our paper according to your comments.

Reviewer 2 Report
Table 2 is missing from the paper. There is a typo on line 232 "Individuals" and 234 delete the word "was". The introduction and conclusion are excellent. The results section need improvement. Table 2 didnt make it into the pdf. I cant see the benefit of presenting the same data in so many different ways. It is confusing. Figure 2 it would seem is just another manipulation of Table 2, I suggest present one or the other as a table or graph. It is not clear if the difference shown n Figure 2 is statistically significant.The difference between "eGFR change" and "eGFR change rate" is not clear and I suspect the difference does not justify the 2 different results in table S1. I think the data taken at only 2 time points makes the results fairly unreliable but this is alluded to in the discussion. Overall a good job with unique data set and interesting (although not convincing to me) results.Author Response
Thank you very much for your valuable comments. We heve revised our paper according to your comments.

Reviewer 3 Report
The manuscript titled “Estimated 24-h urinary sodium-to-potassium ratio is related to renal function decline: A 6-year cohort study of Japanese urban resident” determined the 6-year renal function decline among urban Japanese community dwellers with no history of metabolic syndrome.
The work is well conducted and the manuscript is well prepared.
The introduction is way too short and needs to be expanded for additional background and rationale of the work.
Although statistical analyses were done, they were not indicated in table 1. Also, what is the meaning of parenthesis in table 1? Please describe the abbreviations first time they are mentioned in the abstract, text or table/figure titles. The titles of the tables/figures need to be clearly described with the addition of new sentences.
Authors need to redesign the table 2 as in the current form it’s not understandable, Is there something missing in table 2?
Author Response
Thank you very much for your valuable comments. We have revised our paper according to your comments.

Round 2
Reviewer 1 Report
I would like to congratulate the authors with these nice manuscript after revisions. You have raised the quality of the manuscript significantly.
I have only a minor correction:
Introduction:
1: The authors must address that estimated glomerular filtration rate and proteinuria both are insensitive predictors of future decline in kidney function. Therefore, simple and alternative biomarkers are extremely interesting in these field.
“→ We have agreed with your opinion. We added the following to Limitation at the end of the Discussion. (line 314-318)
Fourth, originally, renal function should be accurately measured using inulin clearance and creatinine clearance, but these are expensive and time-consuming. Although eGFR using CKD-EPI was used in this study, eGFR and proteinuria are insensitive indicators for predicting renal function, and future development of simple and more alternative biomarkers is desired.”
Answer: Maybe I have not been clear enough in round 1: I think you should describe the introduction “eGFR and proteinuria are insensitive indicators for predicting renal function decline, and future development of CKD”. This is the reason why the effect of the sodium-to-potassium ratio (Na/K) is relevant to investigate. I think you should delete the added line 314-318.
Author Response
Thank you very much for your comment.We have revised our paper.
